# Exploring the Inhibitory Effect of AgBiS_2_ Nanoparticles on Influenza Viruses

**DOI:** 10.3390/ijms241210223

**Published:** 2023-06-16

**Authors:** Junlei Yang, Lihuan Yue, Bei Shen, Zhu Yang, Jiang Shao, Yuqing Miao, Ruizhuo Ouyang, Yihong Hu

**Affiliations:** 1Institute of Bismuth and Rhenium Science, School Materials and Chemistry, University of Shanghai for Science and Technology, Shanghai 200093, China; fuyangjunlei@163.com (J.Y.); 203612304@st.usst.edu.cn (Z.Y.); yqmiao@usst.edu.cn (Y.M.); 2Institut Pasteur of Shanghai, Chinese Academy of Sciences, University of Chinese Academy of Sciences, Shanghai 200031, China; 3CAS Key Laboratory of Molecular Virology & Immunology, Institutional Center for Shared Technologies and Facilities, Pathogen Discovery and Big Data Platform, Institut Pasteur of Shanghai, Chinese Academy of Sciences, Shanghai 200031, China; lhyue@ips.ac.cn (L.Y.); bshen@ips.ac.cn (B.S.); 4School of Life Sciences and Biotechnology, Shanghai Jiao Tong University, Shanghai 200240, China; 5Institutional Center for Shared Technologies and Facilities of Institut Pasteur of Shanghai, Chinese Academy of Sciences, Shanghai 200031, China

**Keywords:** nanoparticles, influenza virus, viral inhibition, virus-cell internalization, intracellular replication

## Abstract

Influenza viruses are respiratory pathogens that are major threats to human health. Due to the emergence of drug-resistant strains, the use of traditional anti-influenza drugs has been hindered. Therefore, the development of new antiviral drugs is critical. In this article, AgBiS_2_ nanoparticles were synthesized at room temperature, using the bimetallic properties of the material itself to explore its inhibitory effect on the influenza virus. By comparing the synthesized Bi_2_S_3_ and Ag_2_S nanoparticles, it is found that after adding the silver element, the synthesized AgBiS_2_ nanoparticles have a significantly better inhibitory effect on influenza virus infection than Bi_2_S_3_ and Ag_2_S nanoparticles. Recent studies have shown that the inhibitory effect of AgBiS_2_ nanoparticles on the influenza virus mainly occurs in the stages of influenza virus-cell internalization and intracellular replication. In addition, it is found that AgBiS_2_ nanoparticles also have prominent antiviral properties against α and β coronaviruses, indicating that AgBiS_2_ nanoparticles have significant potential in inhibiting viral activity.

## 1. Introduction

The influenza virus is a kind of respiratory pathogen. After a mild infection, it can cause symptoms, such as runny nose, fever, and cough. In severe cases, it can cause pneumonia, which is one of the major disease endangering the life and safety of the patient. The US Food and Drug Administration (FDA) has only approved two types of drugs to treat influenza viruses, namely M2 protein ion channel blockers and neuraminidase inhibitors [1]. With the long-term use of drugs, drug-resistant strains have emerged which have restricted the use of these two types of anti-influenza virus drugs. This has become a public health problem worldwide. On the other hand, vaccines for influenza viruses often fail as the virus mutates. Therefore, it is necessary to find new anti-influenza virus drugs or broad-spectrum antiviral drugs with different antiviral mechanisms to replace the traditional drugs.

In recent years, nanomaterials have been rapidly developed and are widely used in biotherapy. In addition, with the deepening of research, it has been found that metal nanomaterials have antibacterial [2], anti-tumor [3], anti-virus [4], and anti-parasitic properties [5]. Studies have found that Ag and Au nanoparticles have prominent antiviral properties. Their action occurs before the virus invades the cell and during the process of crossing the cell membrane. However, some things could be improved in the antiviral materials reported at this stage. First of all, the inhibitory effect of nanomaterials is mainly to block virus adsorption and entry into cells, and there is no apparent inhibitory effect in the late stage of virus replication. Secondly, the application of single metal has certain limitations. Therefore, multi-metal nanomaterials are used in antiviral therapy to enrich the characteristics of nanomaterials, improve their antiviral performance, and show excellent antiviral capabilities [6,7].

Researchers have found that bismuth compounds have good photothermal conversion ability in the near-infrared region in tumor treatment, so they can be used as a photothermal treatment of tumor reagents and can achieve ideal therapeutic effects. Regarding antibacterial properties, studies have found that the bismuth ions in metallothionein in organisms can interact strongly with cysteine, providing hope for applying bismuth compounds to inhibit bacteria and viruses [8]. At this stage, bismuth coordination compounds, such as bismuth subsalicylate and ranitidine bismuth citrate, can form a protective film in the stomach, prevent gastric acid and protease from harming the stomach, and inhibit *Helicobacter pylori*. It is widely used to treat gastric ulcers [9,10]. In addition, research has shown that ranitidine bismuth citrate can play an essential inhibitory role in the late stage of the severe acute respiratory syndrome (SARS) virus replication cycle through the coordination of bismuth ions [11]. With the outbreak of the coronavirus disease 2019 (COVID-19), many drugs have been used to restrain or kill the new coronavirus. Studies have shown that bismuth salt inhibits the NTPase and RNA helicase of the new coronavirus type 2 [12]. Moreover, studies have found that ranitidine bismuth citrate, used to treat stomach diseases, can inhibit the replication of new coronavirus type 2 and reduce the symptoms of pneumonia in mice [13].

In addition, Ag also plays a vital role in the field of anti-virus. Park et al. synthesized a kind of magnetic hybrid colloid loaded with Ag nanoparticles of different sizes. Using the interaction between Ag and biological macromolecules, Park explored the inhibitory effect of those nanoparticles on bacteriophages, noroviruses, and adenoviruses. The results show that Ag nanoparticles are combined with sulfhydryl-containing proteins on the virus’s surface, thereby destroying the virus envelope and killing the virus. Therefore, Ag is widely used to prepare antibacterial and antiviral materials due to its good bactericidal and antiviral properties [14].

We synthesized Bi-based nanomaterials and explored their antiviral potential. Their inhibitory effect on influenza virus infection at the cellular level was tested. Nanomaterials were added at different stages of virus-infected cells, and the inhibition effect was evaluated via TCID50, fluorescent quantitative PCR (qRT-PCR) and indirect immunofluorescence methods. Furthermore, the antiviral effect of bimetallic nanomaterial AgBiS_2_ was intensively investigated. Our research may provide simple, green, efficient, and low-cost antiviral nanomaterials for application.

## 2. Results

### 2.1. Material Characterization

Three kinds of nanoparticles (AgBiS_2_, Bi_2_S_3_, and Ag_2_S) are rapidly synthesized at room temperature, and the synthesized nanoparticles are tested and characterized using TEM and XRD (Figure 1). As shown in Figure 1a, the experimentally synthesized AgBiS_2_ material is irregularly granular and crystalline, with an average particle size of 18.6 ± 0.3 nm, uniform particle size, and good dispersion. The XRD test of the experimentally synthesized AgBiS_2_ shows that the XRD diffraction peaks of AgBiS_2_ nanoparticles are consistent with the standard card JCPDF 89-2045 (Figure 1d). These data indicate the successful synthesis of AgBiS_2_ nanoparticles. As shown in Figure 1b, the shape of Bi_2_S_3_ nanoparticles is similar to that of mulberries, with irregular shapes and a particle size of about 40 nm. In addition, XRD analysis shows that the diffraction peaks of the experimentally synthesized Bi_2_S_3_ nanoparticles are consistent with the standard card JCPDF 75-1036 (Figure 1e). As shown in Figure 1c, the experimentally synthesized Ag_2_S nanoparticle is spherical with an average particle size of 64 nm. TEM shows that part of Ag_2_S is connected with multiple morphologies because PVP wraps the outer surface. Additionally, the comparison of the Ag_2_S standard card (JCPDF 89-3840) shows that the XRD diffraction peaks of the test group match the standard card (Figure 1f). In summary, the synthesis of experimental nanomaterials is successful.

In order to verify the successful synthesis of AgBiS_2_ nanoparticles, the synthesized material was subjected to high-angle circular dark scene scanning (mapping). It can be observed from Figure 2 that the distribution of the three elements of Bi, S, and Ag in nanoparticles further demonstrates the successful synthesis of AgBiS_2_ nanoparticles.

After synthesizing the three kinds of nanomaterials, their chemical properties were tested. The analysis shows (Figure 3a) that the hydrodynamic diameters of AgBiS_2_, Bi_2_S_3_, and Ag_2_S are 43.3 nm, 52.1 nm, and 76.3 nm, respectively, based on the hydrated particle size (DLS).

As shown in Figure 3b, the zeta potentials of AgBiS_2_, Bi_2_S_3_, and Ag_2_S in an aqueous solution are −12.56 mV, −8.03 mV, and −20.5 mV, respectively. The zeta potential of AgBiS_2_ is between Bi_2_S_3_ and Ag_2_S. With the addition of Ag, the zeta potential value decreased from −20.5 mV to −12.56 mV. The infrared analysis (IR) of the synthesized nanoparticles and PVP, as shown in Figure 3c, highlights that the stretching vibration peak of the amino group (N-H) of PVP was around 3400 cm^−1^, and that the stretching vibration peak of the carbonyl group (C=O) was at 1685 cm^−1^. The alkyl group (C-H) was stretched and flexed by the nanoparticles at 2871 cm^−1^, the alkyl group (C-H) at 3027 cm^−1^ was the stretching vibration peak, and at 896 cm^−1^ the alkyl group (C-H) was the rocking vibration peak [15,16]. Thermogravimetric analysis (TG) shows that the weight of Bi_2_S_3_ begins to decrease at 200–300 °C (Figure 3d). This stage indicates that the loss of unstable oxygen photo energy groups on the material leads to a decrease in the weight of the material. The decrease in weight at 300–350 °C is attributed to the carbonization of the PVP material wrapped in the outer layer, which separates the PVP wrapped in the outer layer from the Bi_2_S_3_ inside. The decrease in weight at 450–550 °C is attributed to the collapse of the material skeleton. For AgBiS_2_, the 3.46% weight loss at 300–350 °C is lower than that of Bi_2_S_3_ at 5.13%. This indicates that in the synthesized materials, the amount of PVP wrapped on the outer surface of Bi_2_S_3_ is higher than that of AgBiS_2_.

### 2.2. Cytotoxicity Test

Good biological safety is the prerequisite for using nanoparticle in biological therapy. The biocompatibility of synthetic nanoparticle is evaluated by studying the toxicity of three nanoparticles to MDCK cells. As shown in Figure 4, the MTT method detects the cell viability of the three nanoparticles incubating with cells for 24 and 48 h. It is found that when the AgBiS_2_ concentration reaches 150 μg/mL, the survival rate of MDCK cells after 48 h of co-culture is still higher than 90%, indicating that AgBiS_2_ has lower cytotoxicity than Ag_2_S and Bi_2_S_3_. However, when Ag_2_S and Bi_2_S_3_ were co-cultured with cells for 48 h, the cell survival rate was higher than 90%, while the concentration was lower than 100 μg/mL. The above results show that the biological safety of AgBiS_2_ is significantly higher than that of Ag_2_S and Bi_2_S_3_, and that further biological experiments should be carried out within the biological safety concentration of each group of materials.

### 2.3. Antiviral Effect Evaluation Experiment

#### 2.3.1. Direct Inactivating Group

Firstly, the direct killing of the materials against viruses was tested. Different concentrations of nanoparticles and viruses were incubated at room temperature for 1 h. The incubated virus solution was used to infect cells, and then the relative content of the virus was tested via RT-qPCR after 48 h of co-culture [17]. The results are shown in Figure 5. Although AgBiS_2_, Ag_2_S, and Bi_2_S_3_ nanoparticles have weak infection inhibition abilities against the influenza virus, the antiviral effects of these three types are not obvious. The infection inhibition rate against the virus was about 20%. It shows that the three kinds of nanoparticles cannot kill the virus through interaction with the virus.

#### 2.3.2. Co-Cultivation Inhibition Group

In order to verify whether nanoparticles interfere with the process of virus adsorption to cells, nanoparticles and influenza viruses were incubated with the cells for an hour at the same time. After washing off unabsorbed influenza viruses and nanoparticles, the RT-qPCR tests were performed after 48 h of incubation [18]. As shown in Figure 6, the virus suppression effect of AgBiS_2_ and Ag_2_S is higher than the Bi_2_S_3_ group, and the best suppression effect of the Ag_2_S group is 71.45%. Among them, Ag_2_S shows a good antiviral effect at a concentration of 10 μg/mL. The inhibition rate of the influenza virus reached 47.69%, which is significantly higher than 40.4% in the AgBiS_2_ group and 14.79% in the Bi_2_S_3_ group. With the increase in the nanoparticles’ concentration, the viral inhibitory effect is improved and tends to be flat. The above results show that Ag_2_S exhibits the best influenza virus inhibitory effect among the three nanoparticles when the nanoparticles and influenza virus are added to the cell together. The antiviral effect of AgBiS_2_ is the second best. However, Bi_2_S_3_ shows very weak antiviral ability. By comparing the three kinds of nanoparticles, it is found that in the presence of Ag ions, the process of virus adsorption to cells will be significantly interfered, thereby reducing the amount of influenza virus adsorbed to cells and achieving the effect of inhibiting influenza virus infection.

#### 2.3.3. Nanoparticle Preculture Inhibition Group

The preliminary exploration shows that the three kinds of nanoparticles can play a blocking role in the process of virus adsorption into cells. To further determine the antiviral effects of the three nanoparticles in the cell, pre-cultivating the nanoparticles at different concentrations with the cells was carried out for 3 h before incubating with the viruses. After the virus and cells were co-cultured for 1 h, the unabsorbed viruses were washed away. After 48 h of culture, the inhibitory effect of different materials on the influenza virus was determined via RT-qPCR (Figure 7). When the nanoparticles entered the cells earlier than the viruses, the degree of intervention of the nanoparticles on the influenza virus replication inside the cells was examined. As shown in Figure 7a, AgBiS_2_ exhibits a significant inhibitory effect on influenza viruses at low concentrations. As the concentration of AgBiS_2_ increases, the inhibition rate reaches 78.78%. The maximum inhibition rate is significantly higher than 64.24% in the Ag_2_S group and 48.53% in the Bi_2_S_3_ group (Figure 7b,c). It indicates that AgBiS_2_ can interfere with the intracellular replication process of the virus. The reason is that Bi^3+^ and Ag^+^ have an affinity for the biological macromolecular protein of viruses and can bind to the biological protein through coordination or chemical reaction so that the protein molecules can be changed in conformation and lose the biological activity [8,19]. Among them, the virus suppression effect of the bimetallic AgBiS_2_ nanoparticles was significantly better than that of the single metal Ag_2_S and Bi_2_S_3_ nanoparticles.

#### 2.3.4. Virus Post-Infection Inhibition Group

To explore the contribution of nanoparticles’ interference to the transmembrane process of the influenza virus, after the virus was adsorbed on the cell surface, the nanoparticles were added and cultured for 48 h. RT-qPCR shows that adding AgBiS_2_, Ag_2_S, and Bi_2_S_3_ enhances the inhibitory effect of the influenza virus (Figure 8). The comparison of the inhibitory effects of the three nanoparticles revealed that the maximum inhibitory rate of AgBiS_2_ on the influenza virus was 91.72%, which is significantly higher than the inhibitory effects of the Ag_2_S and Bi_2_S_3_ groups. In this virus post-infection group, nanoparticles participate in the virus-cell internalization process in the early stage and the virus intracellular replication process in the later stage. Referring to the results of the material preculture group, AgBiS_2_ and Ag_2_S can interfere with the replication stage of the influenza virus and suppress virus propagation. In the virus preculture group, nanoparticles affect both the transmembrane and virus replication processes in cells. Both AgBiS_2_ and Ag_2_S shows the inhibiting effect on the virus. However, the inhibiting effect of AgBiS_2_ is higher than that of Ag_2_S, which indicates that AgBiS_2_ plays an inhibitory role in the process of virus transmembrane due to the introduction of Bi^3+^. It can also be mutually confirmed by comparing the virus inhibition effects of Bi_2_S_3_ between the virus post-infection inhibition group and the nanoparticle preculture inhibition group.

### 2.4. TCID_50_ Test for Virus Activity

After treatment in each experimental group, the virus-containing supernatant was collected, and the TCID_50_ method was used to detect the virus titer and verify whether the virus in the supernatant was living. After the influenza virus was cultured for six days, the TCID_50_ of the influenza virus was measured via hemagglutination assay.

For AgBiS_2_ nanoparticles, as shown in Figure 9, by comparing the four different inhibition pathways, it was found that after AgBiS_2_ nanoparticles treatment, the influenza virus titer of the group (d) decreases most significantly with 1.58 lg (TCID_50_), followed by 1.1 lg (TCID_50_) of the group (c). It indicates that the main inhibitory effect of AgBiS_2_ nanoparticles on the influenza virus occurs during the cellular internalization and intracellular replication stages. These data further confirm the results of RT-qPCR detection.

For Ag_2_S nanoparticles, the results of the RT-qPCR test show that the inhibitory effect of Ag_2_S on the influenza virus is lower than that of AgBiS_2_. TCID_50_ is used to determine the live virus in the virus-containing supernatant. It is found that the Ag_2_S-group (c) has the best inhibitory effect, and the titer of the influenza virus decreases by 1.14 lg (TCID_50_) (Figure 10), but it is lower than the inhibitory effect of the AgBiS_2_-group (d) (Figure 9). It indicates that the main inhibitory stage of Ag_2_S nanoparticles against influenza viruses occurs in the virus’s intracellular replication process.

For Bi_2_S_3_ nanoparticles, detection of the influenza virus titer of each experimental group shows that the virus titer of the group (d) decreases. It shows an inhibitory effect on the influenza virus at a lower concentration. The inhibitory effect is low for group (a) and group (b) (Figure 11). The above conclusions are consistent with the results of RT-qPCR.

### 2.5. Indirect Immunofluorescence Experiment

The viruses mainly invade cells by the fusion of virus surface proteins and cell surface receptors. After the virus is adsorbed on the cell surface, part of the virus antigen protein will be exposed to the cell’s outer layer. As a standard virologic technique, the indirect immunofluorescence assay (IFA) demonstrates the influenza viral antigens expressed in infected cells. The virus antigen can be detected to indirectly determine the degree of infection of the virus to the cell. At this time, the indirect immunofluorescence method can determine the number of virus-infected cells [19,20,21].

Indirect immunofluorescence has been extensively used as a confirmatory assay in viral research. Here, IFA was performed to study the inhibitory impact of nanoparticles on the expression of influenza virus HA antigen. After pre-cultivating the virus and cells for 1 h, AgBiS_2_ solution at different concentrations (0, 50, and 150 μg/mL) was added, and the cells were fixed on the glass bottom 96-well plate after 24 h. As shown in Figure 12, using a rotating disc confocal microscope it is observed that when the concentration of AgBiS_2_ nanoparticles is 0 μg/mL; the fluorescence intensity of the virus-positive control group was higher. When the concentration of AgBiS_2_ nanoparticles is increased to 50 μg/mL, it is found that the HA intensity reduced significantly compared with the positive control group, indicating that AgBiS_2_ inhibited the process of virus infection. As the concentration of AgBiS_2_ increased to 150 μg/mL, the fluorescence intensity of HA protein further decreased compared with the 50 μg/mL group. It indicates that AgBiS_2_ nanoparticles have an inhibitory effect on influenza virus-infected cells, and as the concentration of the added nanomaterial increases, the inhibitory effect increases.

### 2.6. Inhibitory Effect on Influenza A Virus, α and β Coronaviruses

To verify the broad-spectrum antiviral properties of AgBiS_2_ nanomaterials, we explored the antiviral effect of AgBiS_2_ nanomaterials on influenza virus A, α and β coronaviruses. The virus post-infection inhibition group approach was applied. After culturing the AgBiS_2_ nanomaterials with the infected cells for 48 h, it is observed from Figure 13a that the inhibitory effect on influenza A is significantly increased with the increased concentrations of AgBiS_2_ nanomaterials. The highest inhibition rate at 100 μg/mL was 90.25%. In addition, we verified the infection-inhibitory effect of AgBiS_2_ nanomaterials on β coronavirus OC43 and α coronavirus 229E. Figure 13b,c show that the AgBiS_2_ nanomaterials also have a significant inhibitory effect on coronavirus. As the concentration of AgBiS_2_ nanomaterials increased, the inhibition effect was the highest at 100 μg/mL, 88.25% and 86.06% for β coronavirus OC43 and α coronavirus 229E, respectively. It indicates that AgBiS_2_ nanomaterials have broad-spectrum and excellent antiviral properties.

## 3. Discussion

The research on nanomaterials has become one of the emerging hot spots, greatly expanding its application [22,23]. However, the application of metal nanomaterials was limited due to their poor biological safety for a long time [24]. As the preparation of nanomaterials is improved by adjusting the proportion of raw materials and exploring the appropriate reaction conditions, nanomaterials with small sizes, high specific surface area, adjustable particle size and flexible surface functionalization are developed [25]. Moreover, with the development of metal nanomaterials in anti-tumor research, metal nanomaterials have shown extensive application prospects in the biological field [3,26]. Furthermore, with the recent outbreak of novel coronavirus, researchers are paying attention to treating viral infectious diseases with metal-based nanomaterials [12,13,20].

In this study, we screened the bismuth-based nanomaterials with anti-virus activity, especially the inhibitory effect on the influenza B virus, and explored the possible inhibitory stages. The virus replication was inhibited by PVP-coated AgBiS_2_ nanomaterials, which is quite similar to Hongze Sun’s lab results, wherein the suppressive effect on SARS-CoV-2 by metallodrug ranitidine bismuth citrate was demonstrated [13]. The three inhibition pathways of metal nanomaterials as antiviral therapeutic materials against viral infections, including direct inactivation [14,21,27,28], inhibition of virus adsorption and entry [29,30,31], and intracellular virus suppression [18,19], are very common [32]. However, further study on the mechanism of AgBiS_2_′s antiviral effect will identify the related molecules and pathways.

Pure bismuth and unmodified bismuth compound, or their nano crystallized materials, have poor solubility, so we used the PVP-coated AgBiS_2_ with good biocompatibility which enhanced the nanomaterial’s solubility and its internalization into cells [33,34]. Jiangfeng Du’s work shows that poly(vinylpyrrollidone)- and selenocysteine-modified Bi_2_ Se_3_ nanoparticles (PVP-Bi_2_ Se_3_@Sec NPs) are in vivo, in vitro low toxicity and biodegradable with dramatic biological effect [33]. Similarly, it is indicated that PEGylation effectively reduces cytotoxicity and increases zinc oxide nanoparticles’ antiviral activity against herpes simplex virus type 1 [34]. Meanwhile, a kind of iron oxide nanoparticles (IO-NPs) developed to fight against the H1N1 influenza A virus was evaluated via MTT and TCID50 and proven to have excellent performance [29]. In addition, the functional modification of nanomaterials was carried out based on the characteristics of virus receptors on the cell surface [25,35]. The strategy proved very effective.

We propose that PVP-AgBiS_2_ nanomaterials internalized into cells affect the pivotal enzymes related to virus replication. The enzyme motif analysis may help us speculate the interaction model between bismuth and enzymes. A functional study will be performed to elucidate the mechanism of bismuth’s effect on enzymes in viral replication. Hopefully, the in-depth understanding will helpfully reveal the target of bismuth during its inhibition of virus propagation.

## 4. Materials and Methods

### 4.1. Synthesis of Nanoparticles

#### 4.1.1. Synthesis of Bi_2_S_3_ Nanoparticles

We dissolved 121 mg of Bi(NO_3_)_3_·5H_2_O in 20 mL ethylene glycol, added 300 mg of PVP-k30, and stirred it with ultrasonic until dissolved. In addition, we weighed 20 mg of thiourea, added it to 3 mL of ethylene glycol and 2 mL of ethanol, and dissolved it by ultrasound. Then, we added dropwise the thiourea solution to the Bi(NO_3_)_3_ solution, stirred rapidly for 15 min, centrifuged and washed with ethanol and ultrapure water several times, and then dispersed in PBS (pH = 7.4) for later use.

#### 4.1.2. Synthesis of Ag_2_S Nanoparticles

We weighed 40 mg of AgNO_3_, dissolved it in 20 mL of ethylene glycol, stirred it to dissolve properly, and then added 300 mg of PVP-k30 to dissolve ultrasonically. Another 20 mg of thiourea was dissolved in 3 mL of ethylene glycol and 2 mL of ethanol ultrasonically. Then, this mixture was added to the AgNO_3_ solution and stirred rapidly for 15 min, and the centrifugal washing step was the same as the synthesis of bismuth sulfide above.

#### 4.1.3. Synthesis of AgBiS_2_ Nanoparticles

We took 61 mg of Bi(NO_3_)_3_·5H_2_O and 21 mg of AgNO_3_ to dissolve them in 20 mL of ethylene glycol, and then added 300 mg of PVP-k30 through ultrasonic blending. Then, we added 20 mg of thiourea, and the other operations were the same as the synthesis of bismuth sulfide above. Bi(NO_3_)_3_·5H_2_O, PVP-k30, AgNO_3_, thiourea, ethanol, and ethylene glyco were purchased from Sinopharm Group Chemical reagent Co., LTD., Shanghai, China.

### 4.2. Characterization of Nanoparticles

We observed the morphology and structure of the synthesized nanoparticles through transmission electron microscopy (TEM, HT7800; Hitachi High-Tech, JPN, Tokyo, Japan) and obtained the energy spectrum data of the material through energy dispersion spectroscopy (EDS, Ultra55; Zeiss, GER, Oberkochen, Germany); Use Zeta potential measuring instrument (Nano-zs90; Malvern, UK) to determine the particle size distribution (DLS) and zeta potential of nanoparticles.

### 4.3. Cultivation of Cells and Viruses

Madin–Darby canine kidney (MDCK) cells (ATCC, CCL-34), an immortal human hepatic cell line HuH-7 (JRCB0403) and RD cells (ATCC, CCL-136), preserved by the Institut Pasteur of Shanghai, Chinese Academy of Sciences, were used as host cells for influenza viruses, coronavirus 229E and OC43, respectively. The cells were cultured in EMEM (ATCC, 302003) supplemented with 10% fetal bovine serum (FBS, Gibco™ 10099141C, Thermo Fisher Scientific Inc., Waltham, MA) and 1% penicillin-streptomycin (PS) in an incubator with 5% CO_2_ at 37 °C. Influenza virus B/Mass/3/66 (ATCC, VR-523), influenza A Virus A/WS/33 (ATCC, VR-825), human coronavirus 229E (ATCC, VR-740) and OC43 (ATCC, VR-1558) were deposited by the Institut Pasteur of Shanghai, Chinese Academy of Sciences. The virus titer was determined using the 50% cell culture infectious dose (TCID_50_) endpoint dilution assay, and calculated according to the Reed–Muench formula [36].

### 4.4. Cytotoxicity Test

The biological cytotoxicity of the prepared nanoparticles was evaluated via the MTT assay. First, the MDCK cells were seeded in a 96-well flat-bottom plate at a density of 10^4^ cells/well, and were next cultured overnight in a 37 °C, 5% CO_2_ incubator. Then, the cells grew about 80–90% of the density, aspirated the culture medium in the wells, and then we washed the plates twice with PBS and added 100 μL of serum-free medium to each well of nanoparticles at different concentrations (0, 10, 20, 50, 100, 150, and 200 μg/mL) with a blank control, and applied six replicate wells for each concentration, and placed them in a 37 °C, 5% CO_2_ incubator. After 24 or 48 h, we aspirated and discarded the cell supernatant, washed the plates twice with PBS, and added 100 μL of new serum-free medium. Then, we added 25 μL of MTT (5 μg/mL), put the plates in the incubator for 4 h, added 100 μL of 10% SDS extraction buffer, and measured the absorbance at 570 nm after overnight incubation.

The following formula was used to calculate the nanomaterial’s cytotoxicity (percent coefficient of variation, CV%). CV% $=\frac{ODs-ODm}{ODn-ODm}\times 100\%$. Among them, ODs is the absorbance of the experimental group, ODn is the absorbance of the control group, and ODm is the absorbance of the blank group.

### 4.5. TCID_50_ Assay for Determination of Virus Titer

MDCK/HuH-7/RD cells were cultured overnight in 96-well flat-bottom plates at 2×10^4^ cells/well. Then, we discarded the supernatant, washed the plates twice with PBS, and added a series of virus dilutions in 100 μL of virus infection medium (from 10 to 10^6^ dilutions), ten replicates for each dilution, and two 96-well plates in parallel. Then, we placed the 96-well flat-bottom plates in a 35 °C (for influenza viruses) and 37 °C (for coronaviruses), 5% CO_2_ incubator for 6 days. After that, we transferred 50 μL influenza virus culture supernatant to each well from the 96-well flat-bottom plates to new U-shaped 96-well plates, gently mixed with 50 μL of 1% chicken red blood cells and then recorded the number of non-agglutinate wells after standing for about 30 min. Meanwhile, cytopathic effects (CPE) of coronaviruses were measured using crystal violet staining. The TCID_50_ was calculated according to the Reed–Muench formula, and a titer of 100 TCID_50_/mL was used for in vitro cell experiments.

### 4.6. Virus Inhibition Test

#### 4.6.1. Direct Inactivating Experiment

MDCK cells were cultured overnight in 96-well plates at a density of 2 × 10^4^ cells/well. We used a serum-free medium to prepare the virus with a final concentration of 100 TCID_50_/mL and materials of different concentrations (0, 10, 20, 50, 100, and 150 μg/mL). Then, we mixed the solution, and incubated at room temperature for one hour. Subsequently, we discarded the supernatant, washed the plates twice with PBS, added 100 μL/well of the viral material mixture and incubated the 96-well plates at 35 °C for 1 h. Then, we washed the plates twice with PBS, added 100 μL/well of viral infection base and incubated at 35 °C for 48 h. Finally, the supernatant was collected from each well and the viral RNA extraction from the supernatant was carried out for RT-qPCR and TCID_50_ detection [18].

#### 4.6.2. Co-Cultivation Inhibition Experiment

MDCK cells were cultured overnight in 96-well plates at a density of 2 × 10^4^ cells/well. Then, we used a serum-free medium to prepare 200 TCID_50_/mL virus and nanomaterial dilutions of different concentrations. We discarded the cell supernatant, washed the plate twice with PBS, and added the final concentration of 100 TCID_50_/mL virus mixed with 0, 10, 20, 50, 100, and 150 μg/mL of material diluent. We placed the 96-well plate at 35 °C for 1 h, then washed the plates twice with PBS, added 100 μL/well of viral infection medium and incubated at 35 °C for 48 h. Finally, we collected the supernatant for each well and extracted the viral RNA from the supernatant for RT-qPCR and TCID_50_ detection.

#### 4.6.3. Nanoparticle Preculture Inhibition Experiment

MDCK cells were cultured overnight in 96-well plates at a density of 2 × 10^4^ cells/well. We used a serum-free medium to prepare material dilutions with 0, 10, 20, 50, 100, and 150 μg/mL concentrations. Then, we discarded the cell supernatant, washed the plate twice with PBS, added 100 μL/well of material diluent and incubated it at 37 °C for 3 h. Next, we washed the plates twice with PBS, added 100 μL/well of 100 TCID_50_/mL virus solution and incubated at 35 °C for 1 h. Again, we washed the plates twice with PBS, added 100 μL/well of viral infection medium and incubated at 35 °C for 48 h. Finally, we collected the supernatant for each well and extracted the viral RNA from the supernatant for RT-qPCR and TCID_50_ detection.

#### 4.6.4. Virus Post-Infection Inhibition Experiment

MDCK cells were cultured overnight in 96-well plates at a density of 2 × 10^4^ cells/well. We discarded the cell supernatant, washed the plate twice with PBS, added 100 μL/well of 100 TCID_50_/mL virus solution prepared with a serum-free medium, and incubated at 35 °C for 1 h. Then, we discarded the cell supernatant, washed the plates twice with PBS and add 100 μL/well of material dilutions prepared with viral infections at concentrations of 0, 10, 20, 50, 100, and 150 μg/mL, and incubated at 35 °C for 48 h. Then, we collected the supernatant for each well and extracted the viral RNA from the supernatant for RT-qPCR and TCID_50_ detection, respectively.

### 4.7. Real-Time Fluorescent Quantitative PCR (RT-qPCR) Experiment

In order to determine the RNA content of the virus, a real-time fluorescent quantitative PCR was established. The virus’s total RNA was extracted using a viral RNA extraction kit. We used the one-step fluorescence quantitative PCR kit for RT-qPCR detection. The procedure was as follows: 50 °C, 15 min for reverse transcription; 95 °C, 2 min with 45 cycles of 15 s; and 60 °C, 45 s. The forward primer sequence for the influenza B virus *NS2* gene amplification was 5′-TCCTCAACTCACTCTTCGAGCG-3′, the reverse primer sequence was 5′-CGGTGCTCTTGACCAAATTGG-3′, and the probe sequence was 5′-CCAATTCGAGCAGCTGAAACTGCGGT-3′. The forward primer sequence for the influenza A virus *M1* gene amplification was 5′-GACCRATCCTGTCACCTCTGAC-3′, the reverse primer sequence was 5′-AGGGCATTYTTGACAAAKCGTCTA-3′, and the probe sequence was 5′-TGCAGTCCTCGCTCACTGGGCACG-3′. The forward primer sequence for human coronavirus 229E *N* gene amplification was 5′-TCCTTCCCGGTCTCAGTCG-3′, the reverse primer sequence was 5′-CTGTCACTTGAAGGATTCCGAG-3′, and the probe sequence was 5′-TCGCGGTCGTGGTGAATCCAAACCTCA-3′. The forward primer sequence for human coronavirus OC43 *N* gene amplification was 5′-TCGTTCTGGTAATGGCATCCT-3′, the reverse primer sequence was 5′-CTGATGGTTGCTGAGAGGTAG-3′, and the probe sequence was 5′-CTAAACTGGTCGGACTGATCGGCCCA-3′.

### 4.8. Indirect Immunofluorescence (IFA) Assay

We used IFA to detect the inhibitory effect of nanomaterials on the spread of viral infections and glass-like 96-well plates to culture MDCK cells at a density of 2 × 10^4^ cells/well. The next day, we discarded the cell supernatant, washed twice with PBS, and added 100 μL/well of 100 TCID_50_/mL virus solution, which was prepared with a serum-free medium and incubated at 35 °C for 1 h. Then, we washed the plates twice with PBS, added the AgBiS2 solutions of different concentrations (0, 50, 100, and 150 μg/mL), which were prepared using the virus infection base medium, and cultured at 35 °C for 24 h. Then, we discarded the cell supernatant, added 50 μL/well of 4% paraformaldehyde to fix the cells at room temperature for 5 min, and blocked by adding 1% BSA solution at 4 °C overnight. According to the operating instructions of the influenza B virus (B/Florida/4/2006), we added the primary antibody diluted 1:800 Hemagglutinin/HA Antibody (11053-T62 kit), and then added the secondary antibody (Goat-Anti-Rabbit lgG H&L (Alexa Fluor 488) ab150077) diluted 1:500, finally added DAPI reagent and transferred the plates to an inverted fluorescence microscope for observation.

## 5. Conclusions

In this work, AgBiS_2_ nanoparticles were synthesized by doping with the bimetal method at room temperature. By comparing the synthesized Bi_2_S_3_ and Ag_2_S nanoparticles, the excellent inhibitory ability of AgBiS_2_ nanoparticles on influenza virus B is explored. The cytotoxicity test shows that the three nanoparticles have good biological safety. Then, the inhibitory effect of nanoparticles on influenza virus B was determined via RT-qPCR, TCID_50_, IFA, and other assays. By comparing the inhibitory effect of synthesized Bi_2_S_3_, Ag_2_S and AgBiS_2_ nanoparticles, it is found that the maximum non-toxic concentration of AgBiS_2_ nanoparticles is higher than that of Bi_2_S_3_ and Ag_2_S. In addition, AgBiS_2_ shows an excellent inhibitory effect on influenza virus B, and the inhibitory process occurs at the stages of influenza virus cell internalization and intracellular replication. The optimal inhibition stage of Ag_2_S mainly occurs in the process of virus intracellular replication, while the main inhibition stage of Bi_2_S_3_ to influenza virus occurs in the stage of virus-cell internalization and intracellular replication. It shows that bimetallic nanoparticles have better antiviral effects than single metallic materials. Furthermore, we also verify the excellent antiviral properties of AgBiS_2_ nanomaterials against influenza virus A and coronaviruses. In this paper, bismuth-based nanomaterials have been applied for the first time to inhibit influenza virus infection, and AgBiS_2_ nanoparticles with strong inhibitory effects on influenza virus A, B, and α, β coronavirus have been screened out. Therefore, with future in-depth research, application potential of bismuth-based nanomaterials in the antiviral field could be further demonstrated.

## 6. Patents

The authors declare that there is potential intellectual property regarding the publication of this article.

## Figures and Tables

**Figure 1 ijms-24-10223-f001:**
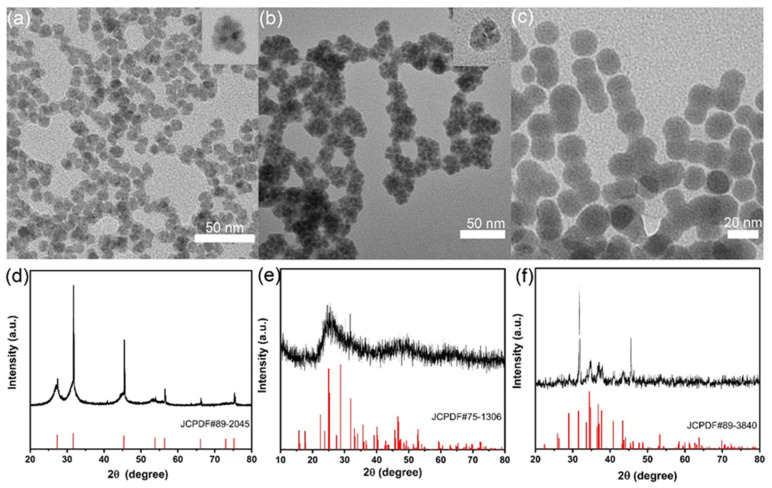
TEM images of nanoparticles (**a**) AgBiS_2_, (**b**) Bi_2_S_3_, and (**c**) Ag_2_S; XRD diffraction patterns of nanoparticles (**d**) AgBiS_2_, (**e**) Bi_2_S_3_, and (**f**) Ag_2_S.

**Figure 2 ijms-24-10223-f002:**
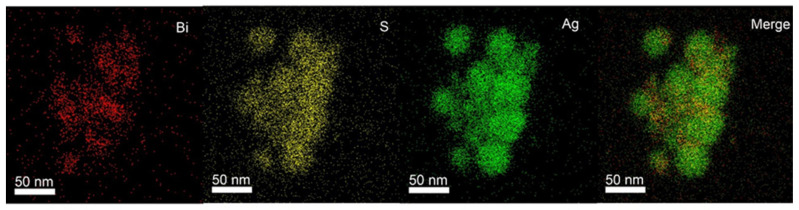
The mapping spectrum of Bi, S, and Ag of AgBiS_2_ nanoparticles.

**Figure 3 ijms-24-10223-f003:**
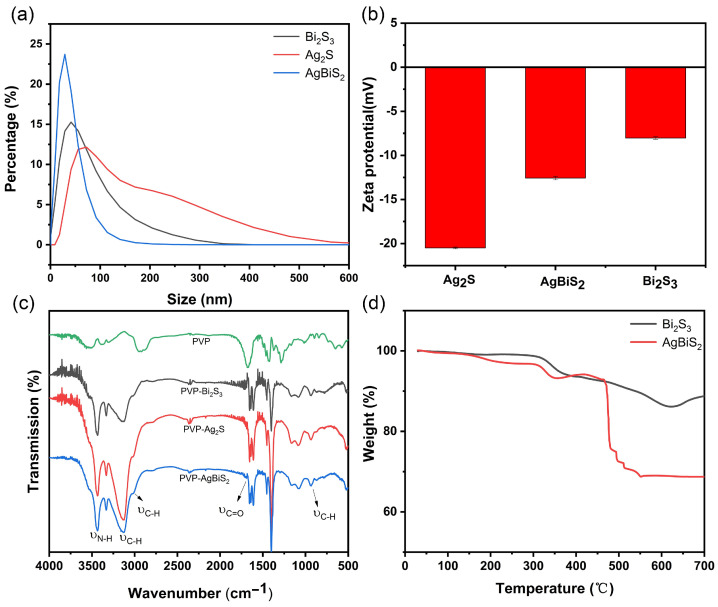
(**a**) Size distributions, (**b**) Zeta potential, (**c**) Infrared spectroscopy analysis of AgBiS_2_, Bi_2_S_3_, and Ag_2_S; (**d**) thermogravimetric analysis of AgBiS_2_, and Bi_2_S_3_.

**Figure 4 ijms-24-10223-f004:**
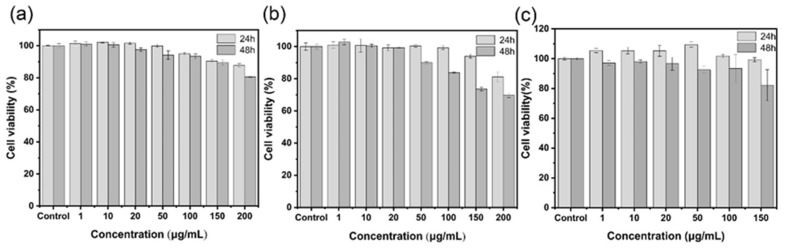
Cytotoxicity tests of (**a**) AgBiS_2_, (**b**) Ag_2_S, and (**c**) Bi_2_S_3_ nanoparticles.

**Figure 5 ijms-24-10223-f005:**
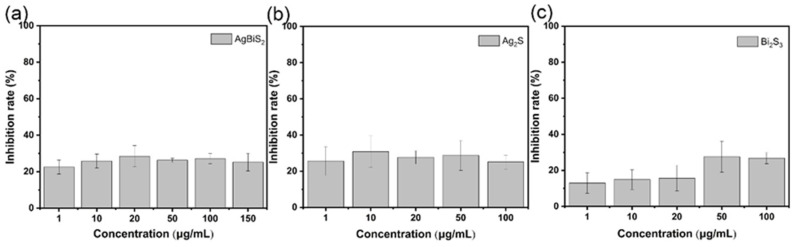
RT-qPCR analysis of (**a**) AgBiS_2_, (**b**) Ag_2_S, and (**c**) Bi_2_S_3_ nanoparticles’ direct inactivating effects on influenza virus B. The inhibitory effect of the control group was 0% which was not shown in the figure.

**Figure 6 ijms-24-10223-f006:**
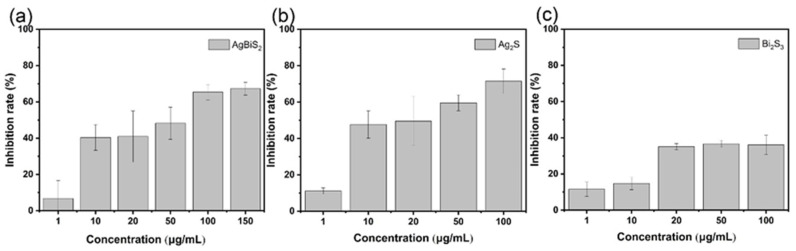
RT-qPCR analysis of (**a**) AgBiS_2_, (**b**) Ag_2_S, and (**c**) Bi_2_S_3_ nanoparticles and influenza virus B co-cultivation of the inhibitory effects. The inhibitory effect of the control group was 0% which was not shown in the figure.

**Figure 7 ijms-24-10223-f007:**
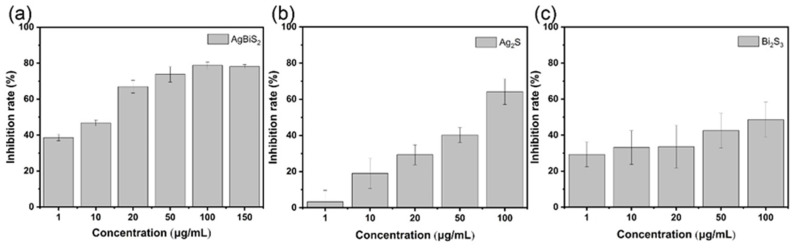
RT-qPCR analysis of the effects on influenza B of the nanoparticle preculture inhibition group. (**a**) AgBiS_2_, (**b**) Ag_2_S, and (**c**) Bi_2_S_3_ nanoparticles. The inhibitory effect of the control group was 0% which was not shown in the figure.

**Figure 8 ijms-24-10223-f008:**
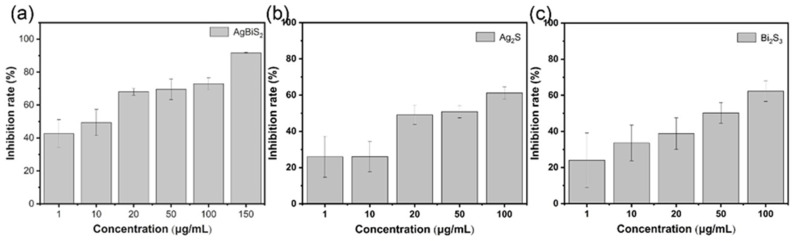
RT-qPCR analysis of the inhibitory effects on influenza B of the virus post-infection inhibition group. (**a**) AgBiS_2_, (**b**) Ag_2_S, and (**c**) Bi_2_S_3_ nanoparticles. The inhibitory effect of the control group was 0% which was not shown in the figure.

**Figure 9 ijms-24-10223-f009:**
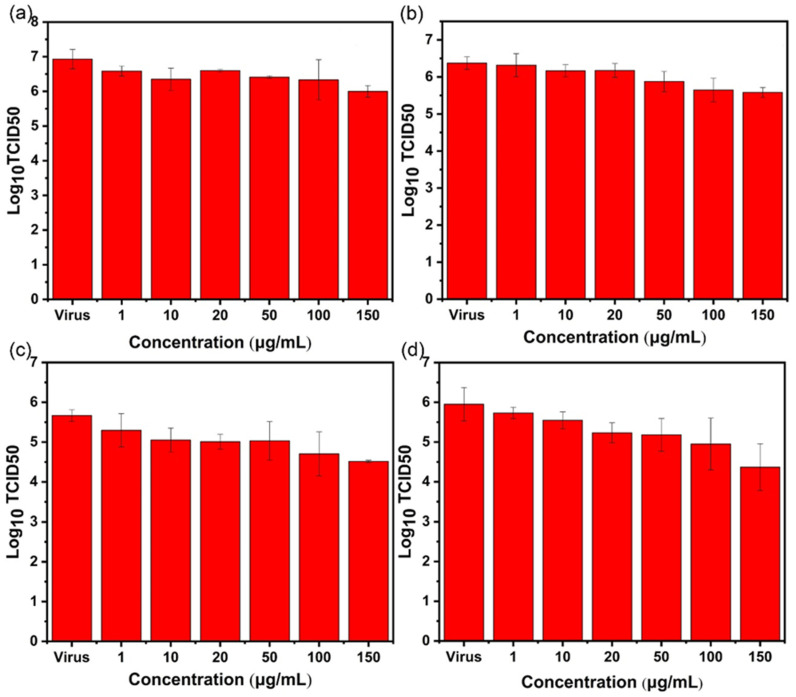
The TCID_50_ method is used to detect the influenza B virus titer of AgBiS_2_ (**a**) direct inactivating group, (**b**) co-cultivation inhibition group, (**c**) nanoparticle preculture inhibition group, and (**d**) virus post-infection inhibition group.

**Figure 10 ijms-24-10223-f010:**
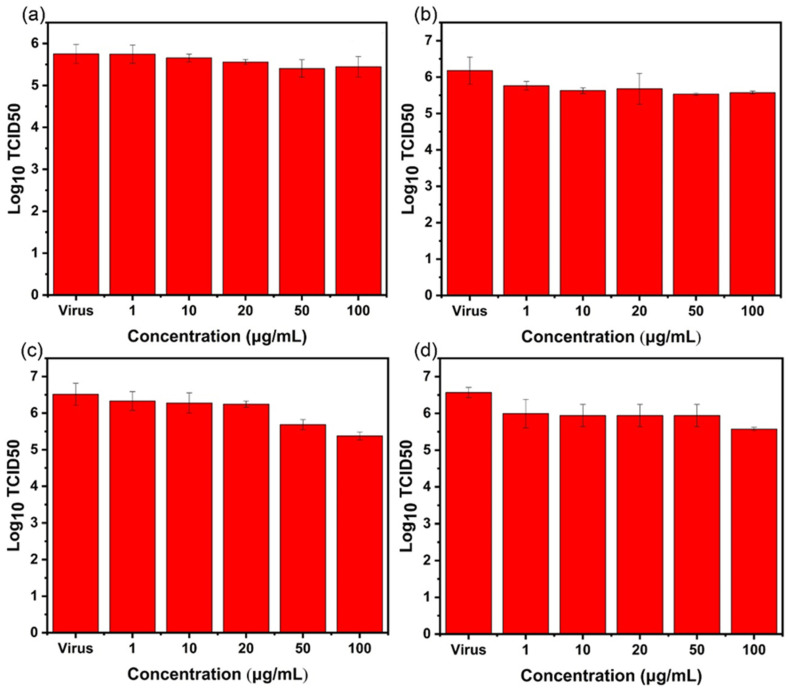
The TCID_50_ method is used to detect the influenza B virus titer of Ag_2_S (**a**) direct inactivating group, (**b**) co-cultivation inhibition group, (**c**) nanoparticle preculture inhibition group, and (**d**) virus post-infection inhibition group.

**Figure 11 ijms-24-10223-f011:**
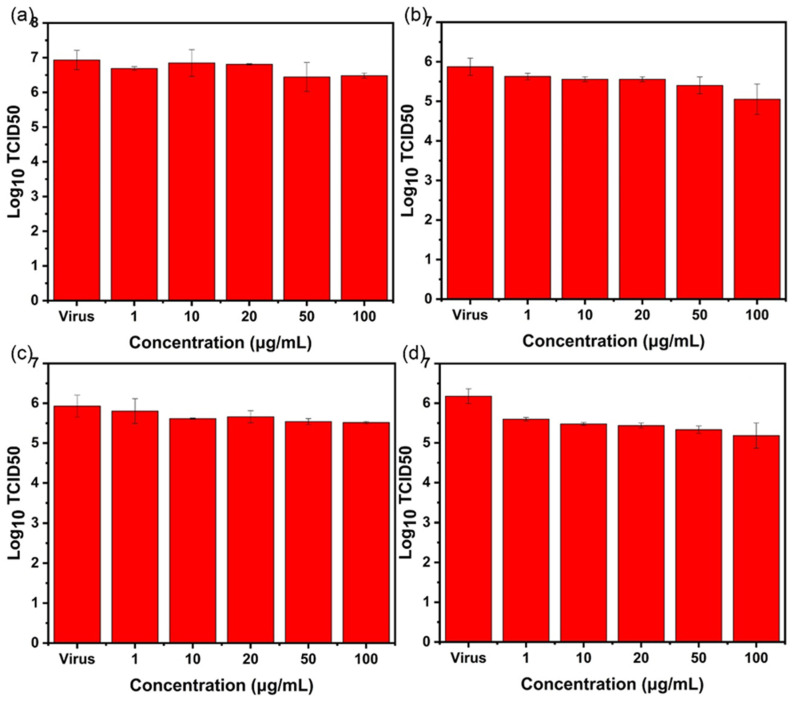
The TCID_50_ method is used to detect the influenza B virus titer of Bi_2_S_3_. (**a**) the direct inactivating group, (**b**) co-cultivation inhibition group, (**c**) nanoparticle preculture inhibition group, and (**d**) virus post-infection inhibition group.

**Figure 12 ijms-24-10223-f012:**
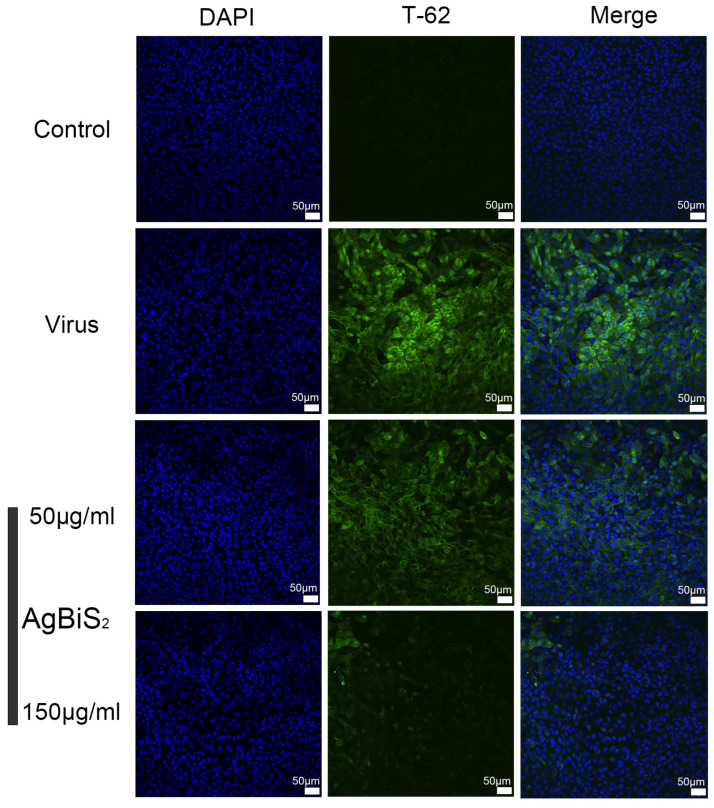
The indirect immunofluorescence assay detects the influenza B virus HA expression affected by AgBiS_2_ at the concentration of 0, 50, and 150 μg/mL, and the virus post-infection group are treated for 24 h. DAPI is a nuclear dye, and T-62 is a fluorescent primary antibody to the influenza virus HA antigen.

**Figure 13 ijms-24-10223-f013:**
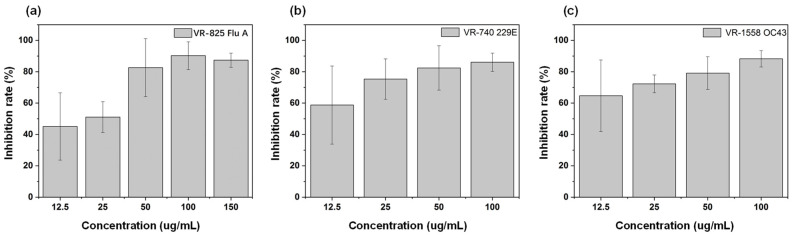
RT-qPCR analysis of the inhibitory effects of the virus post-infection inhibition group on influenza virus A (**a**), β coronavirus OC43 (**b**) and α coronavirus 229E, and (**c**) by AgBiS_2_ nanoparticles.

## Data Availability

The data presented in this study are available on request from the corresponding author.

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
