# Peer review of "Exploring the Inhibitory Effect of AgBiS2 Nanoparticles on Influenza Viruses"

_ijms, 2023, doi:10.3390/ijms241210223_

Round 1

Reviewer 1 Report

Author developed and tested new nanoparticles with anti-viral activity. They found out that adding the silver element increased the inhibition effect of AgBiS2. Author showed that nanoparticles affected the syntheses of the viral/messenger RNA. However, the information about significant inhibitory effect on virus replication is missing. I have same other comments:

-          Experiments fig. 5-8, the control is missing (for example virus preincubated with PBS etc)

-          Fdig.7-12 Information that influenza b virus was used should be included into the legend (text)

-          Can you provide information about the specificity of the primers used in PCR? Which gene they recognized?  

-          Did the author use lower concentration in direct inactivated test? Controls are missing.

-          TCID50 test. 6-day cultivation with influenza virus is very long time. The cells are exhausted after 6 days. The replication cycle of influenza virus is estimated for 8 hours. It means, authors detected the virus released from reinfected cells. Authors could obtain better results after 8-16 h p.i.

-          The hemagglutination test is not providing the information about living and infectious virus. This test detected only HA. The HA test and plaque assay are not always in correlation. You can have very high HA test but very low virus titer obtained by plaque assay.

-          Authors are describing the inhibitory effect of AgBiS2, Ag2S and Bi2S3. The higher concentration of nanoparticles is toxic after 48 h. Here, we are speaking about 6 days. Secondly, authors did not show any statistically decreased of the virus titers. I third, the changes did not depend on the concentration of the tested nanoparticles.

-          Why did authors perform the experiments with influenza viruses at 35°C and not at 37°C?

Reviewer 2 Report

The authors have explored the antiviral activity of AgBiS2 nanoparticles and compared the same with

Ag2S and Bi2S3 nanoparticles against influenza virus A and B, α and β coronaviruses. The methods of synthesis and characterization were well explained, with special mention of the technique of using the mapping spectrum of Bi, S, and Ag of AgBiS2 nanoparticles. The direct inactivating method, co-cultivation inhibition method, and nanoparticle preculture inhibition method were used to determine the antiviral activity of the synthesized nanoparticles using qPCR analysis. An indirect immunofluorescence experiment has also demonstrated the control of influenza virus B in the MDCK cells. The biocompatibility of the synthesized nanoparticles was also assessed. The work is very interesting, but I have some technical comments.

1.      Abstract: I have an objection to the word "first time" report. There are several reports. Therefore, authors can omit the specific term.

2.      Introduction: line 24-25/ Is it the only reason to develop this nanoparticle? If it is, how does the current particle solve this issue? Better to bring other limitations of the standard drugs available for the treatment.

3.      Line 59-60/ Authors are advised to write Park et al., and reference 14 should come at the end of the sentence.

4.      Line 67: 'This research' – what research are you talking about? Are you talking about the present work? I think the last paragraph should speak about the objective of the study, including the outline, not the conclusion.

5.      Result: Line/ 105-106: It's not meaningful. Better to write colloidal size or hydrodynamic diameter.

6.      Line 109-111: the authors have mentioned the sentence, "Ag2S has the highest negative electricity, Bi2S3 has the highest potential, and the potential of AgBiS2 is between Bi2S3 and Ag2S". Can we use the term' negative electricity" to abbreviate zeta potential? Please clarify.

7.      In Figure 3 (a): the ordinate is marked as Percentage (%). The data shows the hydrodynamic diameter of the synthesized nanoparticles, which was plotted as d (nm) along abscissa and Intensity (%) as ordinate. Please rectify the graph. In the figure captions of Figure 3a, it is written as DLS which means the phenomenon because dynamic light scattering (DLS) is a phenomenon. Please change the figure caption. The X-axis should be logarithmic. Please redraw the plot.

8.      In Figure 3 (c), the ordinate is marked as "Transmission". We know that in FTIR, we measure the sample's "transmittance," not "transmission". Please correct the ordinate labeling.

9.      The authors have used Madin-Darby canine kidney (MDCK) cells for assessing the cytotoxicity. It is a canine cell line. What is the justification for using this cell line, and how does it relate to human's biocompatibility to the nanoparticles? Please explain.

10.  In section 2.6, the method for analysis of antiviral activity against influenza virus A, α, and β coronaviruses using AbBiS2 nanoparticles was assessed. It was written that "The Virus post-infection inhibition group approach was applied." Although, in Figure 13 captions, the last sentence mentions "Direct inactivating group, co-cultivation inhibition group, and nanoparticle preculture inhibition group". Please explain the reason for the incorporation of this sentence.

11.  In section 2.5, the sentence "After pre-cultivatint the virus and cells 261 for 1 hour, AgBiS2 solution at different concentrations (0, 50, and 150 μg/mL) were added…….." mentioned a new term "pre-cultivatint". Please explain this new term.

12.  The materials and methods are written as if the authors are commanding the authors to do the experimental steps. It should be totally rephrased using "we have done", or "we have added ", etc. instead of giving instructions to the readers. Please rectify the entire materials and methods section.

13.  Last but not least, the English language needs to be corrected by a professional English-speaking person. The manuscript has a lot of Grammatical mistakes.

14.  Minor revision. 

The topic is interesting but there are several errors. The manuscript may be accepted after a minor revision. 

Round 2

Reviewer 1 Report

I have no other comments.